# Path Loss Prediction Based on Machine Learning Techniques: Principal Component Analysis, Artificial Neural Network, and Gaussian Process

**DOI:** 10.3390/s20071927

**Published:** 2020-03-30

**Authors:** Han-Shin Jo, Chanshin Park, Eunhyoung Lee, Haing Kun Choi, Jaedon Park

**Affiliations:** 1Department of Electronics and Control Engineering, Hanbat National University, Dajeon 34158, Korea; hsjo@hanbat.ac.kr; 2Department of Computer Science, University of Southern California, Los Angeles, CA 90089, USA; chanship@usc.edu; 3Agency for Defense Development, Daejeon 34186, Korea; eunhyoung.lee@add.re.kr; 4TnB Radio Tech., Seoul 08511, Korea; radioeng@tnbrt.com

**Keywords:** wireless sensor network, path loss, machine learning, artificial neural network (ANN), principle component analysis (PCA), Gaussian process, multi-dimensional regression, shadowing, feature selection

## Abstract

Although various linear log-distance path loss models have been developed for wireless sensor networks, advanced models are required to more accurately and flexibly represent the path loss for complex environments. This paper proposes a machine learning framework for modeling path loss using a combination of three key techniques: artificial neural network (ANN)-based multi-dimensional regression, Gaussian process-based variance analysis, and principle component analysis (PCA)-aided feature selection. In general, the measured path loss dataset comprises multiple features such as distance, antenna height, etc. First, PCA is adopted to reduce the number of features of the dataset and simplify the learning model accordingly. ANN then learns the path loss structure from the dataset with reduced dimension, and Gaussian process learns the shadowing effect. Path loss data measured in a suburban area in Korea are employed. We observe that the proposed combined path loss and shadowing model is more accurate and flexible compared to the conventional linear path loss plus log-normal shadowing model.

## 1. Introduction

Wireless sensor network comprises typically multiple sensor nodes and central base stations, where the sensors measure physical status of the environment and report it to the central base station via radio signal. Path loss is a phenomenon between the transmitter and receiver in the strength of radio signal as it propagates through space. Since radio receivers require a certain minimum power (sensitivity) to be able to successfully decode information, path loss prediction is essential in mobile communications network design and planning such as link budget, coverage analysis, and locating base station. Many existing path loss models [1,2,3,4,5,6,7,8,9,10,11,12,13,14] adopt a linear log-distance model, which is empirically derived by assuming a linear proportionality between the path length and the path loss, and by determining a proportional factor through the adequate linear regression analysis of the measured data. The linear log-distance model is simple and tractable, but it does not guarantee accurate path loss prediction performance for all radio propagation environments. Advanced modeling methods are required to more accurately and flexibly represent the path loss for complex and various environments.

Path loss models are classified into empirical, deterministic, and semi-deterministic models. The empirical models are based on the measurements and use statistical characteristics. This model is a simple way of estimating path loss, not taking all of the many parameters that determine path loss into account. Distance is the most essential parameter of path loss model and its physical effect on the path loss is mathematically expressed using a linear log-distance function in literature. Thus, linear log-distance path loss plus shadowing model is used as a baseline for most empirical models and given by [2]
(1)$$PL=PL_{0}+10n\log10\left(\frac{d}{d_{0}}\right)+X_{\sigma },\mathrm{for}\;d\geq d_{0}$$
where *d* is the length of the path and $PL_{0}$ is the path loss at the close-in reference distance $d_{0}$, which is generally determined by the measuring points close to the transmitter. *n* is the path loss exponent representing the rate of change of path loss as the distance increases. The shadow fading denoted by $X_{\sigma }$ is a Gaussian random variable with zero mean and σ standard deviation in dB. Most measurement based path loss models extend the baseline given in Equation (Equation 1) to incorporate the effect of other main radio parameters such as frequency [1,3,4,6], the height of the transmit and receive antennas [1,3,4,6], clutter and terrain [1,5,6], the percentage of the area covered by buildings in the built up area [7], and line-of-sight and non-line-of-sight [8,9,10]. Meanwhile, Jo et al. [14] improved the Modified Hata model to suit the higher frequency band of 3–6 GHz. This model is also an empirical model and adopts a linear log-distance function, which is labeled “(2) Proposed” in Figure 1. It is worth noting that the measured data are well presented by linear regression in Figure 1a, but not especially for the distances less than 200 m in Figure 1b. This is a typical example of the need for a model that can accurately represent both data distributions.

Machine learning is a set of methods based on a dataset and modeling algorithms to make predictions. These days, machine-learning-based techniques are utilized in image recognition, natural language processing, and many other fields. Machine learning tasks can be classified as supervised learning and unsupervised learning. The purpose of supervised learning is to learn a relation or function between inputs and outputs, which is for classification or regression problems. On the other hand, unsupervised learning is to derive hidden rules or relationships from unlabeled data.

The machine learning approach to path loss modeling is expected to provide a better model, which can generalize well the propagation environment since the model is being learned through training with the data collected from the environment. The prediction of propagation path loss is regarded as a regression problem, as stated in the literature [15,16,17]. In this context, path loss models have been developed by various supervised learning techniques such as support vector machine (SVM) [16,18], artificial neural network (ANN) [19,20,21,22], random forest [17], K-nearest neighbors (KNN) [17]. The authors of [19,20,21,22] provided path loss prediction using ANN models, which provide more precise estimation over the empirical models. Zhang et al. [21] developed a real-time channel prediction model, which could predict path loss (PL) and packet drop for dedicated short-range communications.

However, they did not deal with a multi-dimensional regression framework for path loss modeling with various features, called the radio environment factors, such as distance, frequency, antenna height, etc. In regression, many candidate functions are adopted because the related inputs are unknown. However, many of these features would be irrelevant or redundant. Furthermore, most of input features do not provide discrimination ability for prediction. To construct a good estimator, the input data need to be converted into a reduced representation set of features by using dimensionality reduction techniques [23] such as principal component analysis (PCA) or singular value decomposition (SVD). Very few articles (e.g., [16]) have applied PCA to learning path loss with a reduced input data. Meanwhile, no studies have been conducted on the development of shadowing models using machine learning although shadowing is a major factor in determining path loss.

In this work, we employ PCA for extracting key features. Dimensionality reduction is justified from the fact that the actual dimension might be larger than the intrinsic dimension. Then, the objectives of dimensionality reduction are to convert input data into a reduced representation set of features, while keeping as much relevant information as possible. We mainly develop a combined path loss and shadowing model, where ANN multilayer perceptron (ANN-MLP) learns path loss variations dependent on the input data with reduced features. For analyzing shadowing effect, we use Gaussian Process (GP) priors for extracting variance over training data. Based on generalized variance bounds, we can give more accurate confidence level over the path loss mean value. Finally, we attempt to combine ANN and Gaussian process models to build an optimal model in two key aspects: mean path loss value and shadowing effect variance.

The rest of this paper is organized as follows. Section 2 describes procedure of the proposed machine-learning-based path loss analysis. Section 3 presents the simulation results of feature extraction using PCA and its effect on path loss prediction accuracy of ANN-MLP learning. Section 4 and Section 5 detail ANN-MLP and Gaussian process, respectively. Measurement system and scenarios are described in Section 6. Section 7 presents experimental results. We conclude in Section 8.

## 2. Machine Learning Based Path Loss Prediction

The path loss model maps the input features onto the output (path loss observation) values. It is not only important to model a predictor that can produce accurate predictions, but also how derive more generalized model that does fluctuate due to specific conditional inputs. In this paper, as shown in Figure 2, we propose a three-step approach, dimensional reduction techniques, combining nonlinear regression models and variance analysis for predicting the path loss in a suburban environment. While the ANN-MLP-based nonlinear model focuses on how to accurately predict the path loss value, the PCA and the variance analysis balance it to produce a more generalized model.

### 2.1. Feature Selection

Once measured data are collected, a dimensionality reduction strategy is applied to obtain simpler feature space in order to expedite the learning. Although there are various types of features in effect on the path loss variation, practical measurable features are the frequency, the height of the transmitting antenna (elevation altitude), the height of the receiving antenna (elevation altitude), and the difference between these two heights. We analyzed features using PCA for evaluating correlation of values between each features and the path loss. The result shows that log distance and log frequency determine over 70% of the path loss variation. In addition, based on the correlation distribution analysis, it can be inferred that various features are dependent on each other.

### 2.2. Data Preprocessing

ANN is a learning model and its accuracy depends on the training data introduced to it. Aside from its algorithmic and tuning options, well distributed, sufficient, and accurately measured set of data is the prerequisite for acquiring an accurate model. In this perspective, data preprocessing is an essential procedure toward obtaining an ANN learning model. Sampling and normalization are also conducted to reduce process time and prevent bias. The objective of learning is to find the optimal weights on given learning data which enables precise prediction. The key factor for obtaining the right weight is to normalize the magnitude of input values, which minimizes side effects from different scales. For instance, with the same increase with 0.0005, different magnitude of inputs with 0.001 and 0.1 can produce a quite dramatic results in gradient, 0.5 and 0.005. If the input features are not properly normalized, backpropagation with iterative partial derivatives throughout MLP-NN can risk deriving biased weights. Based on propagation characteristics of the input features and balancing the different scale of them, we applied logarithmic transformation on the frequency (MHz) and distance (m) values. Then, data are divided by train and test sets with running iterations by cross-validation for offsetting sampling bias. Cross-validation is a resampling procedure used to evaluate machine learning models on a limited data sample. The procedure has a single parameter called k, which indicates how many groups the given data sample are split into. As such, the procedure is often called k-fold cross-validation. For preparing learning data, all the measured data are divided into two sets, training (80 %) and testing (20 %), with uniform random sampling. The test set is for adjusting hyperparameters for model optimization.

### 2.3. Path Loss Model

ANN is a nonlinear regression system motivated by learning the weighted network of latent variables through backpropagation. The ANN model outperforms the polynomial regression model in prediction performance [24] and handles more dimensions than the look-up table method [25]. Considering the complex propagation due to the fluctuating heights and complex distribution of buildings in urban area, the nonlinear model can fit better to linear regression. We applied the logistic sigmoid function as an activation function within ANN networks.

### 2.4. Shadowing Model

Shadowing is attenuation on signal power, which is caused by obstacles between the transmitter and receiver through scattering, reflection, diffraction, and absorption. The receiver power variation due to path loss is observed over long distances, whereas variation due to shadowing occurs by the formation of obstructing object or the length of it. Traditionally, shadowing was treated as a coefficient with normal distribution since it is hard to generalize the characteristics of obstacles and, generally, its effects on path loss are trivial. However, in outdoor environment with various obstacles, shadowing effects play a significant role in analyzing path loss prediction with certain confidence level.

## 3. PCA

PCA is a dimension reduction technique that linearly transforms the original space into a new space of smaller dimensions while simultaneously describing the variability of the data as much as possible. In fact, the PCA projects along the eigenvectors of the covariance matrix corresponding to the largest eigenvalues, where the eigenvectors points in the direction with the highest amount of data variation. In addition, the magnitude of eigenvalues can be used to estimate the intrinsic dimension of the data. Indeed, if *x* eigenvalues have a magnitude much larger than the remaining ones, the number *x* can be regarded as the true dimension of the data. Clearly, a linear technique is able to only extract linear correlations between variables, whereas it usually overestimates the intrinsic dimension for the data having complex relations.

Figure 3 and Figure 4 show the path loss prediction results of learning with four features (transmitting antenna height, receiving antenna height, transmitting/receiving antennas heights ratio, and distance) and PCA-applied one feature (distance), respectively. We can easily notice that the performance of PCA-applied simple model is similar to the original multi-featured training model, as shown in the summarized prediction loss error comparison in Table 1. The data analysis with major variables from PCA requires more effort in experimental data gathering and learning time cost compared to multiple variables model. PCA can filter out unnecessary noise or less-related independent variables, so that the major variables derived from PCA reflect the more inherent path loss properties and result in higher accuracy. The training cost for applying the learning method is crucial since it can be a feasible factor in a real environment. The one variable model in ANN-MLP and GP model takes 10% and 1%, respectively, of the training time compared to the four-variable model. Therefore, we can focus on a distance feature along the same frequency for ANN model training in the following sections.

## 4. ANN-MLP

The most common type of ANN is the MLP neural network in which multiple neurons are arranged in layers, starting from an input layer, followed by hidden layers, and ending with an output layer. The outputs from each node at the layer are the weighted sum of its inputs over an activation function. We design the fully connected ANN-MLP, which constitutes several hidden layers of nodes and a single hidden layer of the network structure is depicted in Figure 5. More specifically, $z_{n}^{\left(l\right)}={\left[z_{1,n}^{\left(l\right)}\;z_{2,n}^{\left(l\right)}\;\cdots \;z_{M,n}^{\left(l\right)}\right]}^{T}$ is the *n*th column vector of the M×N matrix $Z^{\left(l\right)}$ in the *l*th layers (l=1,2,⋯,L−1), given by
(2)$$\begin{array}{ccc} z_{n}^{\left(l\right)} & = & H\left(a_{n}^{\left(l\right)}\right)={\left[H\left(a_{1,n}^{\left(l\right)}\right)\;H\left(a_{2,n}^{\left(l\right)}\right)\;\cdots \;H\left(a_{M,n}^{\left(l\right)}\right)\right]}^{T}, \end{array}$$
(3)$$\begin{array}{ccc} a_{n}^{\left(l\right)} & = & \left\{\begin{array}{cc} W^{\left(l,n\right)}\cdot x_{n} & \mathrm{for}\;l=1\\ W^{\left(l,n\right)}\cdot z_{n}^{\left(l-1\right)} & \mathrm{for}\;l=2\cdots L-1\;, \end{array}\right. \end{array}$$
where $x_{n}={\left[x_{1,n}\cdots x_{D,n}\right]}^{T}$ is the *n*th column vector of the D×N matrix X in input layer. Here, we assume *D* features with each *N* data. The weight matrices for the layer l=1 and for the layers l=2,3,⋯,L−1 associated with $x_{n}$ are given by
(4)$$W^{\left(1,n\right)}=\left[\begin{array}{cccc} w_{1,1}^{\left(1,n\right)} & w_{1,2}^{\left(1,n\right)} & \cdots & w_{1,D}^{\left(1,n\right)}\\ w_{2,1}^{\left(1,n\right)} & w_{2,2}^{\left(1,n\right)} & \cdots & w_{2,D}^{\left(1,n\right)}\\ \vdots & \vdots & \ddots & \vdots \\ w_{M,1}^{\left(1,n\right)} & w_{M,2}^{\left(1,n\right)} & \cdots & w_{M,D}^{\left(1,n\right)} \end{array}\right]\mathrm{and}$$
(5)$$W^{\left(l,n\right)}=\left[\begin{array}{cccc} w_{1,1}^{\left(l,n\right)} & w_{1,2}^{\left(l,n\right)} & \cdots & w_{1,M}^{\left(l,n\right)}\\ w_{2,1}^{\left(l,n\right)} & w_{2,2}^{\left(l,n\right)} & \cdots & w_{2,M}^{\left(l,n\right)}\\ \vdots & \vdots & \ddots & \vdots \\ w_{M,1}^{\left(l,n\right)} & w_{M,2}^{\left(l,n\right)} & \cdots & w_{M,M}^{\left(l,n\right)} \end{array}\right],$$
respectively. In this study, three types of the commonly used activation functions are evaluated, namely rectifier, logistic sigmoid, and hyperbolic tangent functions. The Rectified Linear Unit (Relu) [26] function, given by Equation (Equation 6), is known for ramp function that allows the model to easily obtain sparse representation.
(6)H(a)=max(0,a).

The logistic sigmoid function is a nonlinear activation function that derives smooth thresholding curve for artificial neural network. The output range of the standard logistic sigmoid function is from 0 to 1, from the following equation:(7)$$H\left(a\right)=\frac{1}{1+e^{-a}}.$$

The drawback of standard logistic sigmoid function is that strongly negative inputs are mapped to near zero, which can cause a neural network to become stuck during training. The hyperbolic tangent function, given by Equation (Equation 8), is a differential nonlinear activation function such the negative inputs are mapped to large negative values and the zero inputs are mapped near zero.
(8)$$H\left(a\right)=\frac{e^{a}-e^{-a}}{e^{a}+e^{-a}}.$$

All these activation functions are bounded, nonlinear, monotonic, and continuously differentiable.The universal approximation theorem [27] shows that a feedforward neural network with three layers and finite number of nodes can approximate any continuous function under moderate assumptions on the activation function in any desired accuracy. However, some highly nonlinear problems need more hidden layers and nodes, since the degree of nonlinearity depends on the number of layers and nodes.

The ANN learning is obtained by updating the weights along the MLP neural network in consecutive iterations of feedforward and backpropagation procedures. Using Equation (Equation 2), the feedforward computation is performed on the following equation:(9)$$\begin{array}{c} z_{n}^{\left(L-1\right)}=H\left(W^{\left(L-1,n\right)}H\left(W^{\left(L-2,n\right)}H\left(\cdots W^{\left(2,n\right)}H\left(W^{\left(1,n\right)}x_{n}\right)\right)\right)\right). \end{array}$$

The prediction value ${\hat{y}}_{n}$ from the final output of feedforward procedure is $a_{1,n}^{\left(L\right)}$, which is linear output of $z_{n}^{L-1}$ and $w^{\left(L\right)}=\left[w_{1,1}^{\left(L\right)}w_{1,2}^{\left(L\right)}\cdots w_{1,M}^{\left(L\right)}\right]$ at the last layer without applying activation function as given by
(10)$${\hat{y}}_{n}=a_{1,n}^{\left(L\right)}=w^{\left(L,n\right)}\cdot z_{n}^{\left(L-1\right)}.$$

The objective of the training is minimize the loss function by adjusting the ANN weights $W={\left\{W^{\left(l,n\right)}\right\}}_{l=1,\cdots ,L\mathrm{and}n=1,\cdots ,N}$. The loss function is given by
(11)$$J\left(W\right)=\frac{1}{N}\sum_{n=1}^{N}|{\hat{y}}_{n}-y_{n}{|}^{2}+\frac{\alpha }{2}\sum_{n=1}^{N}\sum_{l=1}^{L}\left|\right|W^{\left(l,n\right)}{\left|\right|}_{2}^{2},$$
where ${\hat{y}}_{n}$ is prediction value for given weight W, and $y_{n}$ is measured path loss values. $\frac{1}{N}\sum_{n=1}^{N}{\left|{\hat{y}}_{n}-y_{n}\right|}^{2}$ is the mean square error (MSE), $\frac{\alpha }{2}\sum_{n=1}^{N}\sum_{l=1}^{L}\left|\right|W^{\left(l,n\right)}{\left|\right|}_{2}^{2}$ is an L2-regularization term that penalizes the ANN model from overfitting, and α is the magnitude of the invoked penalty.

After feedforward phase, adaptive updates for the weight on each connections are conducted by backpropagation [28]. Starting from initial random weights, the backpropagation is repeatedly updating these weights based on gradient descent of loss function with respect to the weights.
(12)$$\begin{array}{ccc} \frac{\partial J}{\partial w_{j,i}^{\left(l,n\right)}} & = & \frac{\partial J}{\partial a_{j,n}^{\left(l\right)}}\left(z_{i,n}^{\left(l-1\right)}\right)\\ \frac{\partial J}{\partial a_{j,n}^{\left(l\right)}} & = & \left\{\begin{array}{cc} H^{\prime }\left(a_{j,n}^{\left(l\right)}\right)\sum_{k=1}^{M}w_{k,j}^{\left(l+1,n\right)}\frac{\partial J}{\partial a_{k,n}^{\left(l+1\right)}} & \mathrm{for}\;l=1\cdots L-1\\ \frac{\partial J}{\partial a_{1,n}^{\left(l\right)}} & \mathrm{for}\;l=L, \end{array}\right. \end{array}$$
where $H^{\prime }\left(a_{j,n}^{\left(l\right)}\right)=dH/da_{j,n}^{\left(l\right)}$ is the derivative for the corresponding activation function. Finally, the weights are updated as follows.
(13)$$w_{j,i}^{\left(l,n\right)}\leftarrow w_{j,i}^{\left(l,n\right)}-\lambda \frac{\partial J}{\partial w_{j,i}^{\left(l,n\right)}}=w_{j,i}^{\left(l,n\right)}-\lambda \frac{\partial J}{\partial a_{j,n}^{\left(l\right)}}\left(z_{i,n}^{\left(l-1\right)}\right),$$
where λ is the learning rate, the hyperparameter for controlling the step-size in parameter updates. This backward pass propagates from the output layer to previous layers with updating weights for minimizing the loss, as shown in Equation (Equation 14). After finishing backpropagation up to the first layer’s weights, it continues to the next iteration of another feedforward and backpropagation process until the weight values are converged certain tolerance level, which is another hyperparameter determining the model. For backpropagation optimization, the Quasi-Newton method, which iteratively approximates the inverse Hessian with $O(N^{2})$ time complexity, is applied. The Limited-memory Broyden–Fletcher–Goldfarb–Shanno (L-BFGS) [29,30,31] is most practical batch method of the Quasi-Newton algorithms and we use the Scipy version of it.

## 5. Gaussian Process

The Gaussian process is kernel-based fully Bayesian regression algorithm that computes a posterior predictive distributions for new data. The Gaussian process is the extension of multivariate Gaussians, which is useful in modeling collections of real-valued variables by its analytical properties. A Gaussian process is a stochastic process based on a sub-collection of random variables that has a multivariate Gaussian distribution. A collection of random variables {h(x):x∈X} is drawn from a Gaussian process with mean function m(·) and covariance function k(·,·) if any finite set of elements $x_{1},\cdots ,x_{m}\in X$, the associated finite set of random variables $h\left(x_{1}\right),\cdots ,h\left(x_{m}\right)$ have distribution,
(14)$$\begin{array}{c} \left[\begin{array}{c} h(x_{1})\\ \vdots \\ h(x_{m}) \end{array}\right]∼N\left(\begin{array}{c} \left[\begin{array}{c} m(x_{1})\\ \vdots \\ m(x_{m}) \end{array}\right],\left[\begin{array}{ccc} k(x_{1},x_{1}) & \cdots & k(x_{1},x_{m})\\ \vdots & \ddots & \vdots \\ k(x_{1},x_{1}) & \cdots & k(x_{m},x_{m}) \end{array}\right] \end{array}\right). \end{array}$$

The mean expression of the Gaussian process model consists of the mean function m(x) and the covariance function, i.e., kernel $k(x,x^{\prime })$, in the following equation. The final expression is expressed as follow.
(15)h(x)∼GP(m(·),k(·,·))

The mean function and covariance function are aptly named as follows,
(16)m(x)=E(h(x))
(17)$$k\left(x,x^{\prime }\right)=E\left[\left(h\left(x\right)-m\left(x\right)\right)\left(h\left(x^{\prime }\right)-m\left(x^{\prime }\right)\right)\right]$$
for any $x,x^{\prime }\in X$. A function h(·) drawn from a Gaussian process prior is an extremely high-dimensional vector from a multivariate Gaussian. In other words, the Gaussian process provides a method for modeling probability distributions over functions.

Let $S=\{{\left(x^{\left(i\right)},y^{\left(i\right)}\right)}_{i=1}^{m}\}$ be a training set with unknown distribution. In the Gaussian process regression model,
(18)$$y^{\left(i\right)}=h\left(x^{\left(i\right)}\right)+\epsilon^{\left(i\right)},i=1,\ldots ,m$$
where the $\epsilon^{\left(i\right)}$ are noise variables with independent $N(0,\sigma^{2})$ distributions. Similar to Bayesian linear regression, a prior distribution over functions h(·) is assumed that it is regarded as a zero-mean Gaussian process prior,
(19)h(·)∼GP(0,k(·,·))
for some valid covariance function k(·,·).

Let $X_{*}$ be testing points drawn from the same unknown distribution. The marginal distribution over any set of input points belonging to *X* must have a joint multivariate Gaussian distribution
(20)$$\begin{array}{c} \left[\begin{array}{c} h(X)\\ h(X_{*}) \end{array}\right]∼N\left(\begin{array}{c} 0,\left[\begin{array}{cc} K(X,X) & K(X,X_{*})\\ K(X_{*},X) & K(X_{*},X_{*}) \end{array}\right] \end{array}\right). \end{array}$$

Based on noise assumption ($\epsilon^{\left(i\right)}∼N\left(0,1\right)$), the noise term can be derived as
(21)$$\begin{array}{c} \left[\begin{array}{c} \epsilon \\ \epsilon_{*} \end{array}\right]∼N\left(\begin{array}{c} 0,\left[\begin{array}{cc} \epsilon^{2}I & 0\\ 0 & \epsilon^{2}I \end{array}\right] \end{array}\right) \end{array}$$

As a result, the sum of independent Gaussian random variables is also Gaussian, thus
(22)$$\begin{array}{c} \left[\begin{array}{c} y(X)\\ y(X_{*}) \end{array}\right]=\left[\begin{array}{c} h(X)\\ h(X_{*}) \end{array}\right]+\left[\begin{array}{c} \epsilon \\ \epsilon_{*} \end{array}\right]∼N\left(\begin{array}{c} 0,\left[\begin{array}{cc} K\left(X,X\right)+\epsilon^{2}I & K(X,X_{*})\\ K(X_{*},X) & K\left(X_{*},X_{*}\right)+\epsilon^{2}I \end{array}\right] \end{array}\right) \end{array}$$

Finally, using the rules for conditioning Gaussians, we have
(23)$$y_{*}|y,X,X_{*}∼N\left(\mu^{*},\Sigma^{*}\right),$$
where $\mu^{*}$ is predicted mean values and $\Sigma^{*}$ is standard deviation values for given distribution, which are given by
(24)$$\begin{array}{lll} \mu^{*} & = & K\left(X_{*},X\right){\left(K\left(X,X\right)+\sigma^{2}I\right)}^{-1}y \end{array}$$
(25)$$\begin{array}{ccc} \Sigma^{*} & = & K\left(X_{*},X_{*}\right)+\sigma^{2}I-K\left(X_{*},X\right){\left(K\left(X,X\right)+\sigma^{2}I\right)}^{-1}K\left(X,X_{*}\right). \end{array}$$

Gaussian process regression derives the model based on the relationship between input (X) and output (Y) being given by the unknown function h. The input data X are the feature values such as distance, frequency, etc. The output data Y are the measured path loss value for given input data. Through the Gaussian process, we learn the relationship h(·) between input and output.

We can estimate the posterior distribution using Bayesian inference, which consists of a prior distribution and likelihood function. Gaussian process is an attractive model for use in regression problems since it allows quantifying uncertainty of predictions due to the errors in the parameter estimation as well as inherent noise in the problem. Furthermore, the model selection and hyperparameter selection methods used in the Bayesian method are directly applicable to Gaussian processes. Similar to locally-weighted linear regression, Gaussian process regression is non-parametric and consequently can model intrinsically arbitrary functions of the input points.

The Gaussian process uses the kernel as a function of covariance that can model various distributions in terms of mean square derivatives for a variety of different data. That is, when x and x’ are close values, the covariance value between those points becomes large, and, inversely, when x and x’ are distant values, the covariance value becomes small. In other words, the closer the value is, the larger the weight it gets, and vice versa. In addition, the distribution of the near and far values is smoothly derived during Gaussian process learning.
(26)$$\begin{array}{c} k\left(x_{i},x_{j}\right)=\exp\left(-\frac{1}{2d{\left(\frac{x_{i}}{length\_scale},\frac{x_{j}}{length\_scale}\right)}^{2}}\right) \end{array}$$

Kernels (also called *covariance functions* in the context of Gaussian processes) are a crucial ingredient of Gaussian processes which determine the shape of the prior and posterior of the Gaussian process. They express the function being learned by defining the *similarity* of two data points combined with the assumption that similar data points have similar target values. The kernel can be divided into stationary and non-stationary kernels. Stationary kernels only depend on the distance of two data points and not on their absolute values, therefore they do not change in the the conversion of input space. On the other hand, non-stationary kernels depend on the specific value of the data point. Stationary kernels can further be classified into isotropic and anisotropic kernels, where isotropic kernels are also unaltered by a rotation of input space. Common types of kernel function are Constant kernel, Radial-basis function (RBF) kernel, Matern kernel, Rational quadratic kernel, Exp-Sine-Squared kernel, and Dot-Product kernel. Alpha value is a value to adjust the noise level, and it is added to the diagonal values of the kernel matrix. The larger is the noise, the more training data should be applied.

## 6. Measurement System and Scenarios

Path loss was measured for three frequencies 450, 1450, and 2300 MHz. The measurement system is composed of a transmitter and a receiver, as shown in Figure 6. The transmitter consisted of cw-generator, power amp, transmitting antenna, and RF cable for connecting each system component. The receiver was composed of the continuous wave (CW) signal receiving part and the GPS signal receiving part. The CW signal receiving part was composed of a receiving antenna, a low noise amplifier (LNA), a spectrum analyzer, and a laptop. The receiving antenna, LNA, and spectrum analyzer were connected by RF cables, and the spectrum analyzer and the laptop were connected by GPIB cable. Both transmitting and receiving antennas had omni-directional radiation patterns. The GPS signal processing system was equipped to obtain accurate distance data between the transmitter and receiver. The GPS signal receiving part consisted of a GPS receiving antenna, a GPS signal processing board, and a laptop. Signals from the GPS signal processing board are transferred to the laptop through a serial port using the RS-232 protocol. The GPS data and CW signal power levels were synchronized using the measured time information, and GPS data were converted into position data.

Measurements were conducted in the suburbs of a small town called Nonsan, which consists of residential areas with 1–4-story buildings, agricultural land, and very low hills. As shown in Figure 7, the transmitting system was installed on the roof of a four-story building, and the transmitting antenna height was 15 m above ground level. The receiving system was mounted on a vehicle and the receiving antenna height was 2 m above ground level. Figure 8 depicts the measurement path and received power level using a color bar. The distance between the transmitter and receiver was within about 3.5 km and the vehicle was moved to receive the power at various distances. The moving speed of the vehicle was maintained so that the separation distance between the measuring points ranged from 25 to 30 cm.

To reduce the Doppler effect due to vehicle movement, the vehicle speed was kept at 30 km/h or less. In the data refining process, the power level measure at 30 km/h or more was filtered out, and then the data were time-averaged to remove the effect of short-term fading. Since the bandwidth used for the measurement was 10 kHz, received power was ideally −134 dBm; however, the noise level observed during field measurements was around −110 dBm. Therefore, −100 dBm, which is above 10 dB of noise signal, was defined as an effective reception level for stable data refinement.

## 7. Results and Discussion

### 7.1. Evaluation Metrics

This section describes experimental results for the different models in three frequencies. The path loss values were predicted with distance from 1 to 3.1 km (in logarithmic distance 0.0–0.49). The predicted path loss values were calculated with the measured data for evaluating prediction error. We applied various prediction error metrics such as root mean squared error (RMSE), mean absolute error (MAE), mean absolute percentage error (MAPE), mean squared logarithmic error (MSLE), and squared R ($R^{2}$) with different perspective. RMSE is defined by
(27)$$\mathrm{RMSE}=\sqrt{\frac{1}{N}\sum_{n=1}^{N}{\left|{\hat{y}}_{n}-y_{n}\right|}^{2}}.$$

RMSE represents the square root of the differences between predicted values and observed values or the quadratic mean of these differences. Although RMSE and MSE are similar in terms of models scoring, they are not always immediately interchangeable for gradient based methods.

MAE is defined by
(28)$$\mathrm{MAE}=\frac{1}{N}\sum_{n=1}^{N}\left|{\hat{y}}_{n}-y_{n}\right|.$$

MAE is a linear index, which means that all the individual differences are weighted equally in the average. It tends to be less sensitive to outliers than MSE.

MAPE is defined by
(29)$$\mathrm{MAPE}=\frac{100\%}{N}\sum_{n=1}^{N}\left|\frac{{\hat{y}}_{n}-y_{n}}{y_{n}}\right|.$$

MAPE is considered as the weighted version of MAE and therefore the optimal constant predictions for MAPE were found to be the weighted median of the target values [32]. MAPE is commonly used as a loss function for regression problems and in model evaluation, because of its very intuitive interpretation in terms of relative error.

MSLE is just MSE calculated in logarithmic scale as
(30)$$\begin{array}{ccc} \mathrm{MSLE} & = & \frac{1}{N}\sum_{n=1}^{N}{\left(log\left(y_{n}+1\right)-log\left({\hat{y}}_{n}+1\right)\right)}^{2}. \end{array}$$
(31)$$\begin{array}{cc} = & MSE(log\left(y_{n}+1\right),log\left({\hat{y}}_{n}+1\right)). \end{array}$$

From the perspective of logarithm, it is always better to predict more than the same amount less than target in that RMSLE penalizes an under-predicted estimate greater than an over-predicted estimate.

$R^{2}$ is closely related to MSE, but has the advantage of no scale. Thus, $R^{2}$ has a value between −*∞* and 1, regardless of the output value. To make it more clear, this baseline MSE can be regarded as the MSE that the simplest possible model would get. The simplest possible model would be to always predict the average of all samples. A value close to one indicates a model with little error, and a value close to zero represents a model very close to the baseline. Thus, $R^{2}$ represents how good our model is against the naive mean model.
(32)$$R^{2}=1-\frac{\sum_{n=1}^{n}{\left(y_{n}-{\hat{y}}_{n}\right)}^{2}}{\sum_{n=1}^{n}{\left(y_{n}-{\bar{y}}_{n}\right)}^{2}}\;,\;\mathrm{where}\;\bar{y}=\frac{1}{n}\sum_{n=1}^{n}y_{n}.$$

### 7.2. Activation Functions for ANN-MLP Model

A key element in the ANN configuration is the activation function that determines the nonlinear transformation for the given learning data. To find the optimal activation function, we examined the prediction loss values (RMSE, MAE, MAPE, MSLE, and R2) which are processed with the validation set, which was initially sampled separately from learning data. To minimize the variance from hyperparameters in learning ANN-MLP models, L-BFGS algorithm was mainly used, which is a batch type of computational optimization method, different from other stochastic mini-batch approaches. For the reference, the fixed hyperparameter of learning rate, epoch, and tolerance rate were set to 0.001, 1000, and 0.00001, respectively, throughout the course of experiments.

For ANN models, to derive a better fit for various distribution and express nonlinearity of path loss characteristics, choosing the right activation function is crucial. Based on the results in Figure 9, Figure 10 and Figure 11, we can find quite diverse figures from alternating activation function network. In Table 2, we evaluate three different activation functions for ANN-MLP network architecture. Comparing hyperbolic tangent and ReLU activation function, the performance of the sigmoid shows better performance in overall metrics. Even though many different types of activation functions exist, we designated the aigmoid function for simplifying network architecture and further performance comparison.

### 7.3. Prediction Accuracy of Path Loss Model

We compared the prediction loss values (RMSE, MAEm MAPE, MSLE, and R2) of the proposed ANN-MLP with other models. The ANN-MLP model is presented in Figure 12, and the popular log linear regression model and two-ray model [33] are presented in Figure 13 and Figure 14. Received signal consists of the line-of-sight path and the ground reflected path in the two-ray model and the resulting path loss is given by
(33)$$\begin{array}{c} PL=\left\{\begin{array}{cc} PL_{c}+40\log_{10}\left(d/d_{c}\right) & \mathrm{for}\;d\geq d_{c}\\ PL_{c}+20\log_{10}\left(d/d_{c}\right) & \mathrm{for}\;d<d_{c} \end{array}\right., \end{array}$$
where break point distance $d_{c}=4h_{t}h_{r}/\lambda$ and the path loss at the distance $PL_{c}=20\log_{10}\left(16\pi h_{t}h_{r}/\lambda^{2}\right)$ are given as a function of transmitting antenna height $h_{t}$, receiving antenna height $h_{r}$, and wavelength λ. Given that the break point distances are 180, 580, and 920 m for frequencies 450, 1450, and 2300 MHz, respectively, and $h_{t}=15\;m$ and $h_{r}=2\;m$, the path loss values beyond the break point distance are shown in Figure 14.

The following sections consider the results of modeling with Gaussian processes shown in Figure 15. Gaussian processes can be modeled for mean values and standard deviations. The mean value is a model of the path loss prediction value, and the standard deviation predicts the range of the expected path loss values. The standard deviation can be used to predict the minimum and maximum path loss that can be distributed at the given input. The Gaussian process was trained using RBF kernel function as
(34)$$\begin{array}{c} k\left(x_{i},x_{j}\right)=exp\left(-\frac{1}{2}d{\left(\frac{x_{i}}{l},\frac{x_{j}}{l}\right)}^{2}\right), \end{array}$$

The RBF kernel is a stationary kernel and also known as the *squared exponential kernel*. It is characterized by a length-scale parameter *l*, which is either a scalar or a vector with the same number of dimensions as the inputs. This kernel is infinitely differentiable, which implies that Gaussian processes with this kernel as a covariance function have mean square derivatives of all orders, and are thus very smooth.

In Table 3, we evaluate the results obtained for linear, ANN-MLP, and Gaussian process model. The prediction loss values (RMSE, MAE, MAPE, and MSLE) from Gaussian process and linear model are higher than ANN-MLP model and the lower $R^{2}$ value shows that its prediction values are not as good fits as ANN-MLP model, which means ANN-MLP slightly outperforms Gaussian process in the path loss prediction. Based on the results in Figure 12, Figure 13, Figure 14 and Figure 15, we can easily find that ANN-MLP model is more sensitive and tweak well on complex data distribution. On the other hand, the straight line is too simple to represent the change in pathloss over distance, and Gaussian process model also shows smooth shapes reluctant to tweak on localized distribution. Unlike other models, the two-ray model for $d\geq d_{c}$ are independent of frequency, as shown in Equation (Equation 34). The two-ray model produces much prediction error compared with other models since its assumption of line-of-sight sight and flat terrain are not guaranteed.

The first Fresnel radius at the midpoint between the transceiver is given by $r=\sqrt{\left(\frac{\lambda d}{4}\right)}$, which range from 6 to 24 m. Given the antenna heights $h_{t}=15\;m$ and $h_{r}=2\;m$, the terrain and objects are likely to be inside of the first Fresnel zone. However, the places for measurement campaign is not ideal flat terrain, but they have different ground height for each receiver position. Accordingly, the clearances of the first Fresnel zone are highly variable. As a result, ANN is more practical in such environments.

### 7.4. Prediction Accuracy of Combined Path Loss and Shadowing Model

As shown in the following equation, a simple log-distance model based on existing deterministic regression analysis has to rely on predetermined variables $PL_{0}$ and *n*, and empirically measured log-normal shadowing $X_{\sigma }$.
(35)$$PL=PL_{0}+10n\log_{10}\left(d\right)+X_{\sigma }\;,\;for\;d\geq 1\mathrm{km},$$
where $PL_{0}$ is the path loss at 1 km distance. On the other hand, the learning-based model can predict the targeted values based on actual distribution of training data, which is able to build more accurate forms of regression model with nonlinear characteristics. In particular, steep slope or complex shaped data distribution can be accurately estimated in this model, since nonlinear model can fit better to those cases than linear model. In addition, the key variables are determined by PCA by evaluating impact level upon co-distribution of input features. By analyzing two variables, the proposed model comprises of ANN-MLP based path loss model and Gaussian process based shadowing model as
(36)$$PL=PL\_ANN\left(d\right)+GP_{\sigma }\;,\;for\;d\geq 1\mathrm{km}$$

In Figure 16 and Table 4, we assess *prediction coverage* that quantifies how closely the predicted values from a model match the actual test data within the standard deviation range. More specifically, the prediction coverage is a ratio of the number of measured data in the range of α to β to total number of measured data, where α and β are given by
(37)$$\begin{array}{c} \alpha =\left\{\begin{array}{cc} PL_{0}+10n\log_{10}\left(d\right)-r\sigma_{1} & \mathrm{for}\;\mathrm{the}\;\mathrm{linear}\;\mathrm{model}\;\mathrm{in}\;\mathrm{Equation}\;(35)\\ PL\_ANN\left(d\right)-r\sigma_{2} & \mathrm{for}\;\mathrm{the}\;\mathrm{proposed}\;\mathrm{model}\;\mathrm{in}\;\mathrm{Equation}\;(36) \end{array}\right. \end{array}$$
(38)$$\begin{array}{c} \beta =\left\{\begin{array}{cc} PL_{0}+10n\log_{10}\left(d\right)+r\sigma_{1} & \mathrm{for}\;\mathrm{the}\;\mathrm{linear}\;\mathrm{model}\;\mathrm{in}\;\mathrm{Equation}\;(35)\\ PL\_ANN\left(d\right)+r\sigma_{2} & \mathrm{for}\;\mathrm{the}\;\mathrm{proposed}\;\mathrm{model}\;\mathrm{in}\;\mathrm{Equation}\;(36) \end{array}\right. \end{array}$$

The conventional shadowing effect is a simple assumption that the mean difference along the predicted value forms the Gaussian distribution, whereas the deviation value derived by the Gaussian Process is based on the correlation analysis. It is a mathematical model for correlation analysis so that errors can be accurately derived. As can be seen from the results in Table 3, the shadowing effect of the existing linear model (1450 MHz, σ: 81.5%, 2σ: 99.9%, 3σ: 100%) shows different coverage than the theoretical value of Gaussian distribution. On the other hand, the predictive model coverage derived from the GP model reflects the actual Gaussian distribution (based on 1450 MHz, σ: 68.4%, 2σ: 95.0%, 3σ: 99.2%). The two models have ideal coverage values of 68%, 95%, and 99.7% for r=1,2,3 when they perfectly match the measure data. For overall frequencies, the proposed model shows around 69.6%, 94.8%, and 99.2% of coverage, whereas the linear model marks 80%, 98%, and 99.8% of coverage for r=1,2,3, respectively. These results mean the proposed model predicts shadowed path loss more accurately than the linear model.

## 8. Conclusions

In this paper, we develop a new approach for the prediction of the path loss based on machine learning techniques: dimensionality reduction, ANN-MLP, and Gaussian process. Tests were designed to evaluate whether dimensionality reduction could support the same path loss prediction accuracy as well as provide confidence level in regression problem based on combining ANN-MLP and Gaussian Process. Using PCA method, dimensionality reduction of the four feature model led to more generalized model with one feature and reduced significant amount of time in training model. The one variable model in ANN-MLP and GP model takes 10% and 1%, respectively, of the training time compared to the four-variable model. By deriving variance from GP, we can design a data-driven model for shadowing effect. The combined path loss and shadowing model, the final outcome of this study, accurately predicts the measured path loss with coverage error less than 1.6 %, whereas the conventional linear log-distance plus log-normal shadowing model has coverage error less than 12%. The new approach would be beneficial for the site-specific design of wireless sensor network with high reliability.

We could extend this study for various ANN structures with different numbers of hidden layers and nodes. Future extension could also develop the ANN-based model that is able to predict only the extra path loss above the reference distance. This makes it possible to develop a model for any reference distance, which will further improve the practicality of the model. It could be possible to apply the proposed method to the development of predictive models with the influence of other variables such as antenna height and building height.

## Figures and Tables

**Figure 1 sensors-20-01927-f001:**
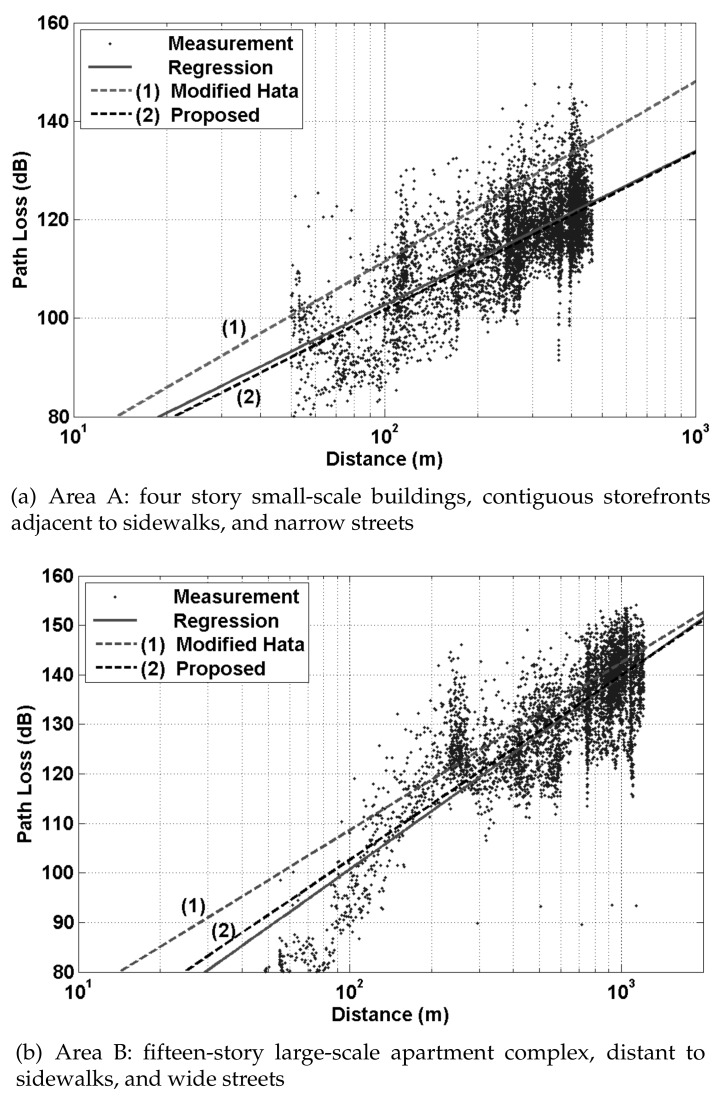
Measured path loss and linear-log distance models presented in [14].

**Figure 2 sensors-20-01927-f002:**
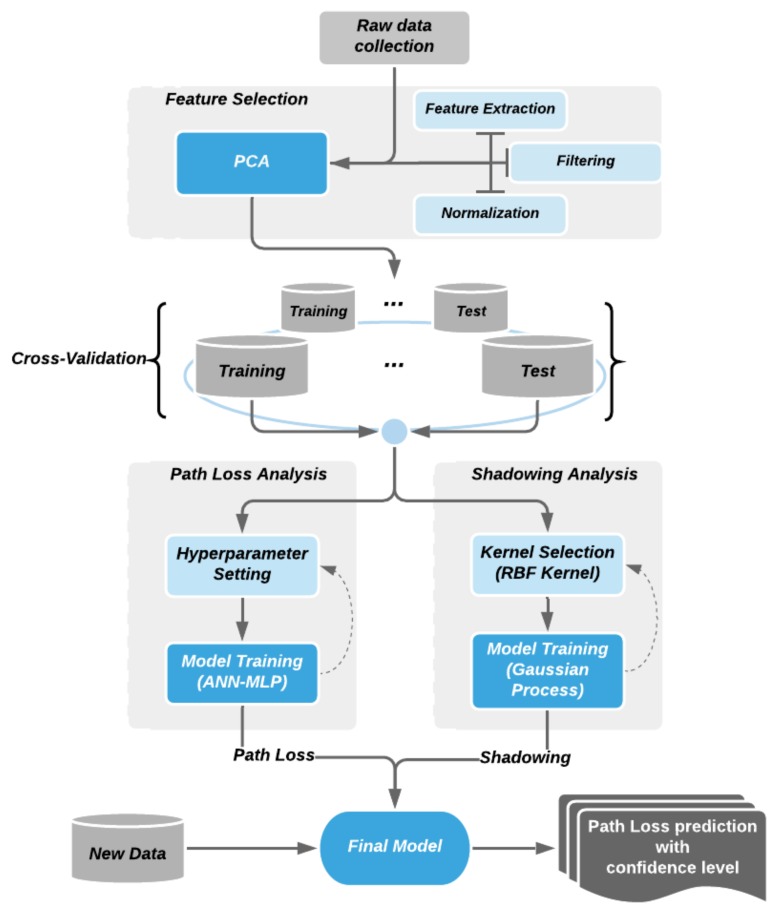
Procedure of machine-learning-based path loss analysis.

**Figure 3 sensors-20-01927-f003:**
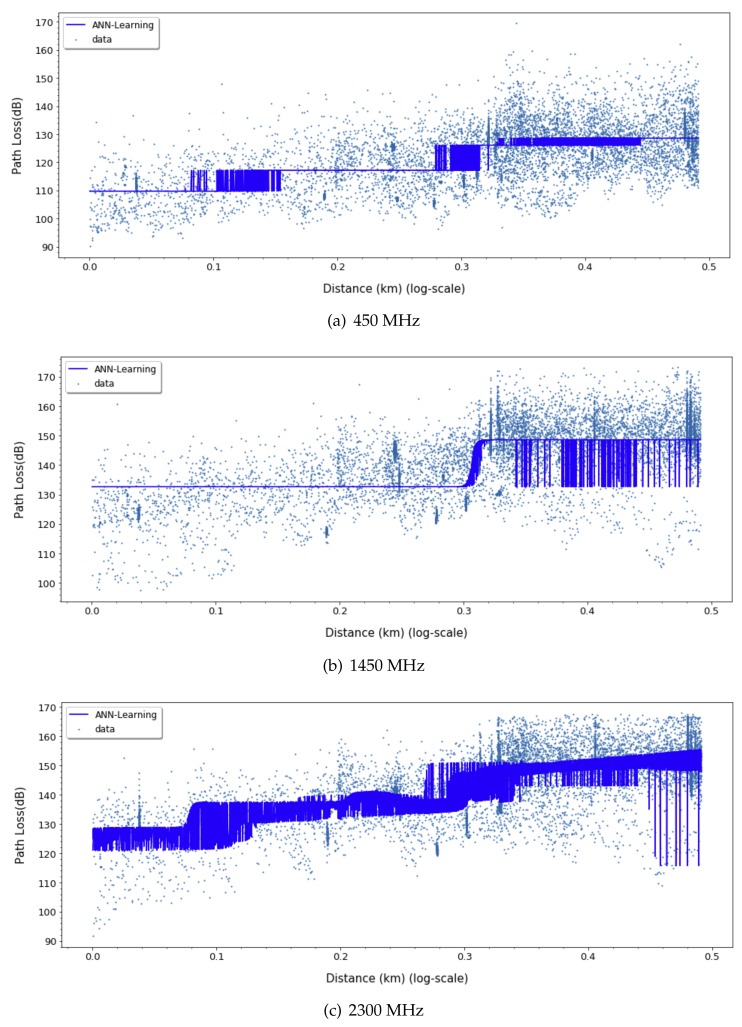
Four-feature-based ANN-MLP model.

**Figure 4 sensors-20-01927-f004:**
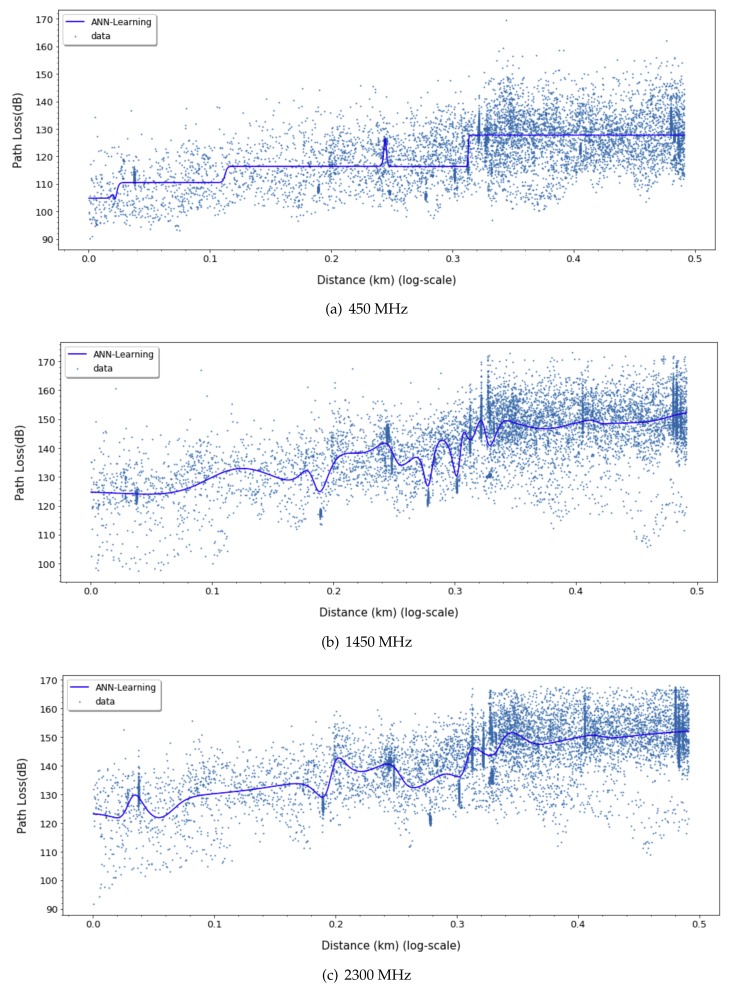
Single-feature-based ANN-MLP model.

**Figure 5 sensors-20-01927-f005:**
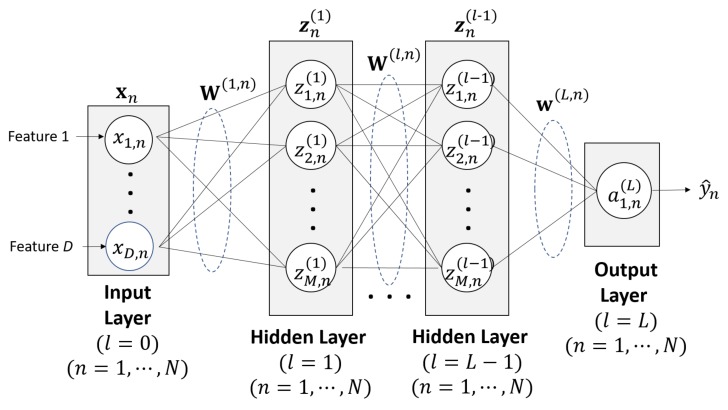
Block diagram of ANN-MLP with *D* features.

**Figure 6 sensors-20-01927-f006:**
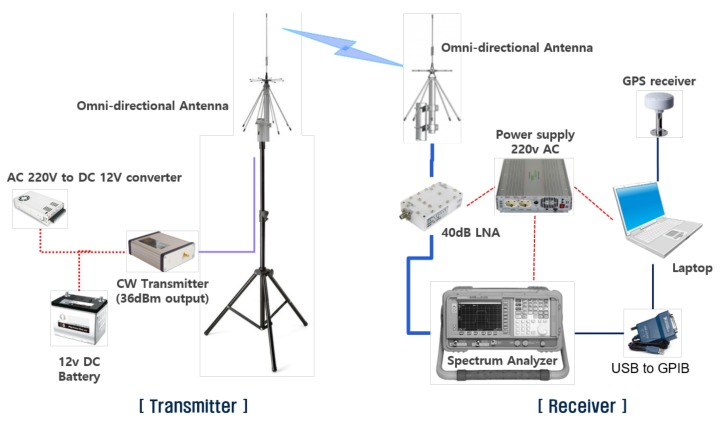
Block diagram of the measurement system.

**Figure 7 sensors-20-01927-f007:**
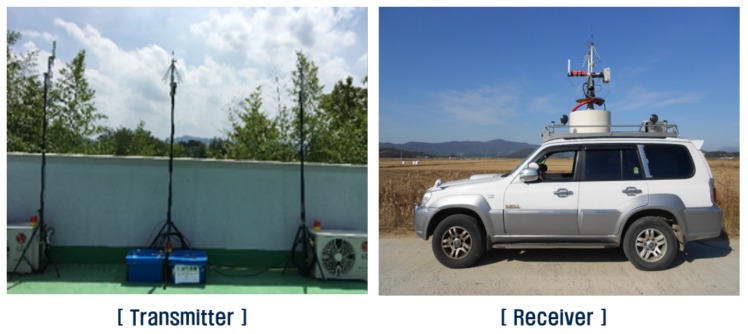
Installation environment of the measurement system.

**Figure 8 sensors-20-01927-f008:**
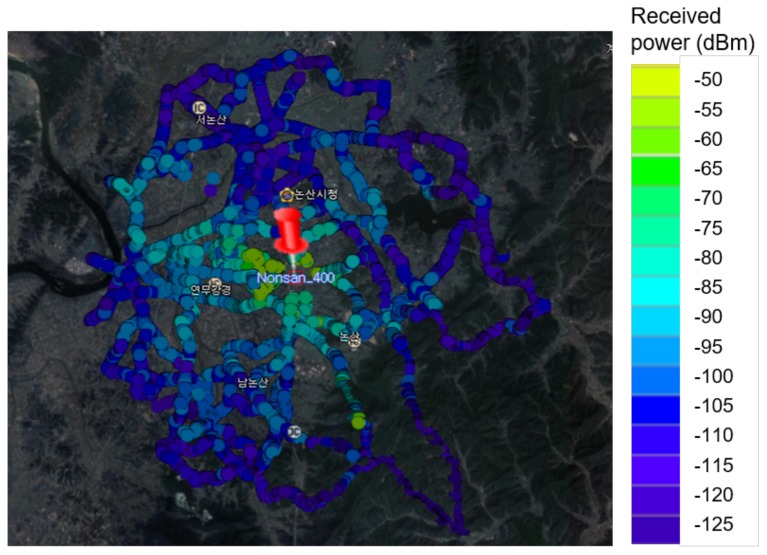
Measurement path and received power level.

**Figure 9 sensors-20-01927-f009:**
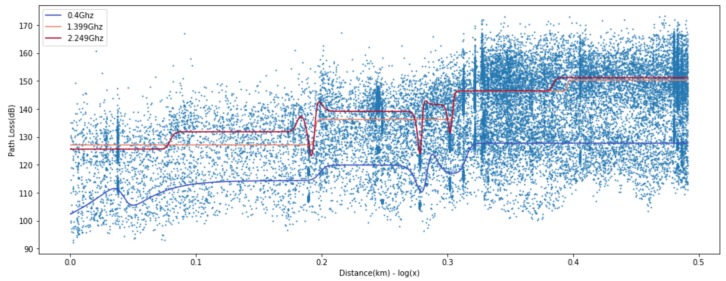
Path loss prediction based on ANN-MLP with Sigmoid activation function for 450, 1450, and 2300 MHz.

**Figure 10 sensors-20-01927-f010:**
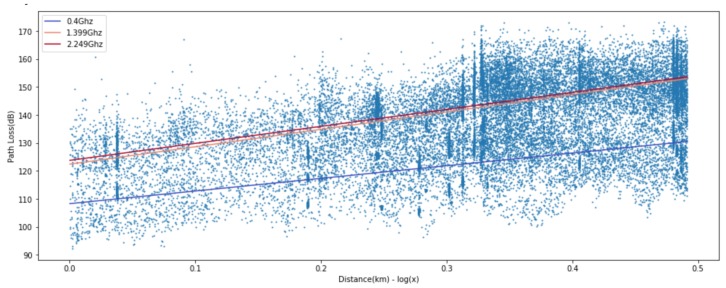
Path Loss prediction based on ANN-MLP with ReLU activation function for 450, 1450, and 2300 MHz.

**Figure 11 sensors-20-01927-f011:**
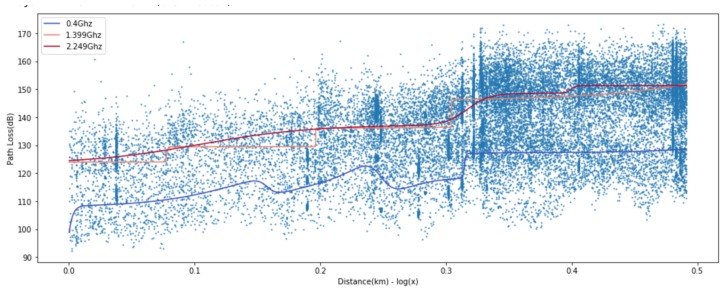
Path Loss prediction based on ANN-MLP with tangent hyperbolic activation function for 450, 1450, and 2300 MHz.

**Figure 12 sensors-20-01927-f012:**
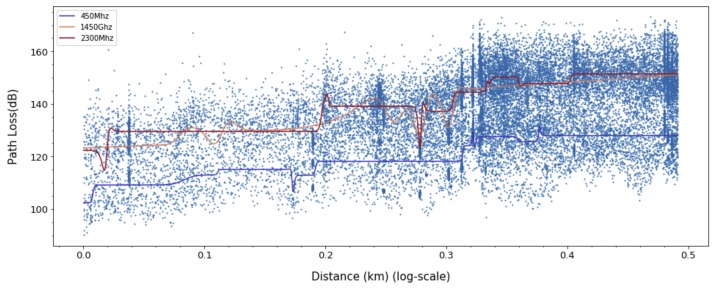
Path loss prediction based on ANN-MLP for 450, 1450, and 2300 MHz.

**Figure 13 sensors-20-01927-f013:**
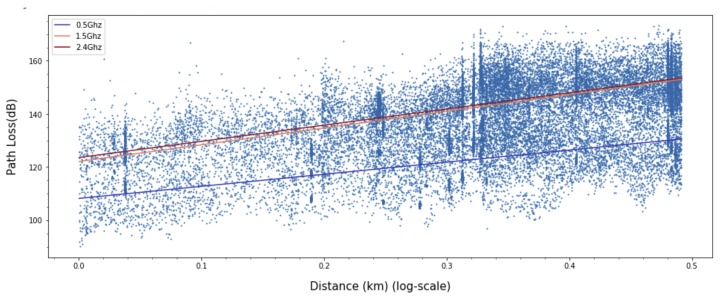
Path loss prediction based on linear model.

**Figure 14 sensors-20-01927-f014:**
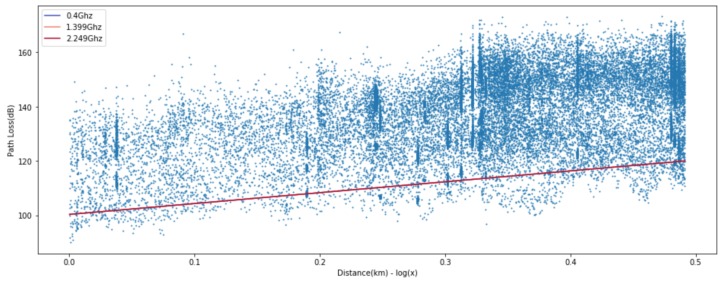
Path loss prediction based on two-ray model.

**Figure 15 sensors-20-01927-f015:**
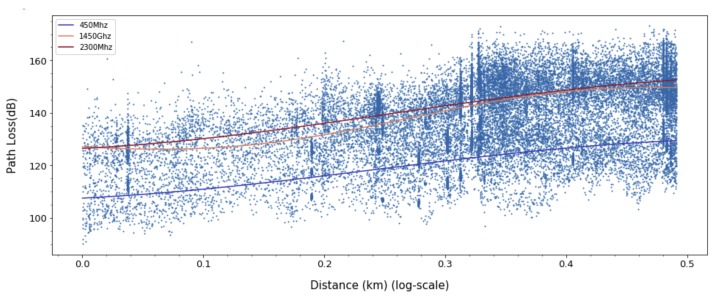
Path loss prediction based on Gaussian process for 450, 1450, and 2300 MHz.

**Figure 16 sensors-20-01927-f016:**
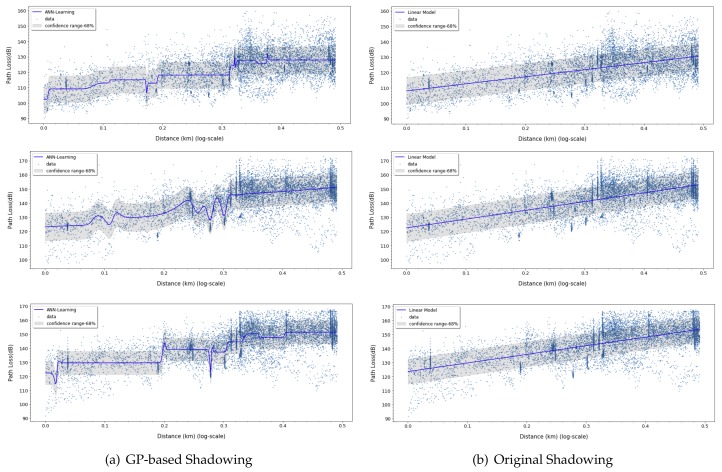
Training graph for the proposed and the linear model with 1× standard deviation.

**Table 1 sensors-20-01927-t001:** Prediction loss comparison for PCA-applied and original 4-featured data.

Model/Error (dB)	RMSE	MAE	MAPE	MSLE	R2
4-feature-based Model	8.40197	6.47590	4.85529	0.00393	0.67463
Single-feature-based Model	8.61307	6.58675	4.94076	0.00413	0.66113

**Table 2 sensors-20-01927-t002:** Comparison of prediction accuracy of different metrics for Gaussian process model.

Model	Error (dB)	RMSE	MAE	MAPE	MSLE	R2
ANN-MLP (Sigmoid)	450 MHz	7.87647	5.89603	4.84501	0.00407	0.39747
	1450 MHz	8.96238	6.80976	4.93310	0.00419	0.46041
	2300 MHz	8.23471	6.20676	4.41919	0.00343	0.48938
	Overall	8.52169	6.48413	4.87674	0.00405	0.66402
ANN-MLP (ReLU)	450 MHz	8.42747	6.84639	5.63144	0.00468	0.31022
	1450 MHz	9.32876	7.22502	5.22711	0.00453	0.41540
	2300 MHz	8.79632	6.75497	4.83320	0.00394	0.41735
	Overall	9.17173	7.23891	5.43902	0.00466	0.61081
ANN-MLP (tanh)	450 MHz	7.93312	6.02763	4.95792	0.00414	0.38877
	1450 MHz	8.90661	6.77473	4.90238	0.00412	0.46711
	2300 MHz	8.54053	6.58882	4.72276	0.00374	0.45075
	Overall	8.54226	6.54968	4.93007	0.00407	0.66239

**Table 3 sensors-20-01927-t003:** Comparison of prediction accuracy of different metrics for Gaussian process model.

Model	Error (dB)	RMSE	MAE	MAPE	MSLE	R2
ANN-MLP	450 MHz	7.87647	5.89603	4.84501	0.00407	0.39747
	1450 MHz	8.96238	6.80976	4.93310	0.00419	0.46041
	2300 MHz	8.23471	6.20676	4.41919	0.00343	0.48938
	Overall	8.52169	6.48413	4.87674	0.00405	0.66402
Linear Model	450 MHz	8.52682	6.93022	5.68143	0.00476	0.31199
	1450 MHz	9.27597	7.19697	5.17684	0.00442	0.41033
	2300 MHz	8.98500	6.96090	4.95213	0.00406	0.38408
	Overall	9.22838	7.30487	5.46315	0.00467	0.60944
Two-ray Model	450 MHz	12.88614	10.64310	8.31157	0.01084	−0.58098
	1450 MHz	30.69274	29.15810	20.03675	0.05465	−5.45023
	2300 MHz	31.50410	30.23772	20.66583	0.05734	−6.86500
	Overall	26.46132	23.34631	16.33805	0.04094	−2.24508
Gaussian Process	450 MHz	8.53235	6.82668	5.60452	0.00479	0.29261
	1450 MHz	9.30737	7.05678	5.11159	0.00447	0.40428
	2300 MHz	8.73691	6.83405	4.85415	0.00387	0.44244
	Overall	8.94361	6.97418	5.23819	0.00446	0.63462

**Table 4 sensors-20-01927-t004:** Prediction coverage by different models.

Frequency	450 MHz	1450 MHz	2300 MHz	Overall
Linear Model (r=1/68%)	76.5%	81.5%	82.1%	80%
Linear Model (r=2/95%)	95.4%	99.9%	100.0%	98%
Linear Model (r=3/99.7%)	99.6%	100.0%	100.0%	99.8%
Proposed Model (r=1/68%)	68.8%	68.4%	71.7%	69.6%
Proposed Model (r=2/95%)	94.2%	95.0%	95.2%	94.8%
Proposed Model (r=3 /99.7%)	99.4%	99.2%	99.2%	99.2%

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
