# Peer review of "Path Loss Prediction Based on Machine Learning Techniques: Principal Component Analysis, Artificial Neural Network, and Gaussian Process"

_sensors, 2020, doi:10.3390/s20071927_

Round 1

Reviewer 1 Report

In this paper, authors focused on the study of path loss prediction using machine learning techniques. Basically, the motivation of the paper is solid as well as the proposed scheme is interesting, novel. Also, the paper has clear contributions technically to relevant research branch. Authors evaluated the proposed approach through extensive experiments with practical settings and scenarios and gave detailed discussion of the experiments results. Moreover, as a whole, the organization and the presented quality of the paper is good.

To improve the paper, the below minor parts are recommended to be modified by authors.
-It is recommended that authors will investigate a few more critical related works and add those works to the manuscript. In particular, authors are able to related studies which were published in MDPI Sensors.
- Authors can describe future research challenges, issues and work for potential researchers in the manuscript.
- Authors need to check if there are any typos, inconsistent parts in the manuscript before publication.

Author Response

Thank you for your comments and suggestions. Please find the attachment.

Reviewer 2 Report

The authors studies ANN based path loss model based on experimental data, which are solid work. However, there are a few issues need to be addressed before it is accepted

1) It is not clear what is the gain to apply ANN to model path loss. The claimed gain of accuracy can be easily done using many other methods, such as curve fitting. The over-fitting of ANN based path loss model could be a issue that prevent presenting a general path loss model

2) It is not clear why and how PCA is adopted to achieve significant results except of log distance and log frequency. The conclusions are trivia based on current knowledge

3) So does the Gaussian processing for shadowing. It is treated in class textbook. Nothing new at all.

Please carefully address your conclusions and contributions. The authors extend the work for various ANN, CNN, LSTM as well as comparing with ARIMA, curve fitting and other typical methods.   

Author Response

(The authors gave the same response as above.)

Reviewer 3 Report

A comparison with the 2-ray model should be also provided. The frequency 450MHz for sure will show a closer fitting to the 2-ray model than to a polynomial regression model.

The receiving antenna height cannot ensure a first Fresnel ellipsoid not obstructed. The grade of obstruction will make that the regression model is not polynomial.

What is the influence of the receiving antenna height on the proposed approach?

Indicate the turning point distance.

More info about 2-ray model can be found on:

  1. Valles Castro, et al., “Measuring the Radiation Pattern of On-Board Antennas at Sea”, Measurement, vol. 111, pp. 167-172,December 2017. DOI: 10.1016/j.measurement.2017.07.042

Author Response

(The authors gave the same response as above.)

Reviewer 4 Report

The paper proposes path loss predictions using machine learning. It is a timely topic, fairly well researched. The results presented appears reasonable. Major comments I have for the authors are as follows:

 1. There is no comparison with other ANN-based path loss predictions in the literature. Therefore, it is hard to quantify the benefits of this approach versus the other ones in the literature. This should be addressed

2. Equation (32), and the ones following it, are valid only when you consider d0=1 m. Otherwise it should be PL= PL0+ 10nlog10(d/d0) + X_sigma. 

3.  I encourage the authors to present some cases where the ANN-based path loss prediction is able to predict just the extra path loss above the reference distance d0. This helps to confirm the utility of the proposed method given different scenarios for d0.  

Author Response

(The authors gave the same response as above.)
